# Robust Data-Driven Dynamic Programming

**Grani A. Hanasusanto**
Imperial College London
London SW7 2AZ, UK
g.hanasusanto11@imperial.ac.uk

**Daniel Kuhn**
École Polytechnique Fédérale de Lausanne
CH-1015 Lausanne, Switzerland
daniel.kuhn@epfl.ch

## Abstract

In stochastic optimal control the distribution of the exogenous noise is typically unknown and must be inferred from limited data before dynamic programming (DP)-based solution schemes can be applied. If the conditional expectations in the DP recursions are estimated via kernel regression, however, the historical sample paths enter the solution procedure directly as they determine the evaluation points of the cost-to-go functions. The resulting data-driven DP scheme is asymptotically consistent and admits an efficient computational solution when combined with parametric value function approximations. If training data is sparse, however, the estimated cost-to-go functions display a high variability and an optimistic bias, while the corresponding control policies perform poorly in out-of-sample tests. To mitigate these small sample effects, we propose a robust data-driven DP scheme, which replaces the expectations in the DP recursions with worst-case expectations over a set of distributions close to the best estimate. We show that the arising min-max problems in the DP recursions reduce to tractable conic programs. We also demonstrate that the proposed robust DP algorithm dominates various non-robust schemes in out-of-sample tests across several application domains.

## 1 Introduction

We consider a stochastic optimal control problem in discrete time with continuous state and action spaces. At any time $t$ the state of the underlying system has two components. The *endogenous* state $\boldsymbol{s}_t \in \mathbb{R}^{d_1}$ captures all decision-dependent information, while the *exogenous* state $\boldsymbol{\xi}_t \in \mathbb{R}^{d_2}$ captures the external random disturbances. Conditional on $(\boldsymbol{s}_t, \boldsymbol{\xi}_t)$ the decision maker chooses a control action $\boldsymbol{u}_t \in \mathcal{U}_t \subseteq \mathbb{R}^m$ and incurs a cost $c_t(\boldsymbol{s}_t, \boldsymbol{\xi}_t, \boldsymbol{u}_t)$. From time $t$ to $t+1$ the system then migrates to a new state $(\boldsymbol{s}_{t+1}, \boldsymbol{\xi}_{t+1})$. Without much loss of generality we assume that the endogenous state obeys the recursion $\boldsymbol{s}_{t+1} = g_t(\boldsymbol{s}_t, \boldsymbol{u}_t, \boldsymbol{\xi}_{t+1})$, while the evolution of the exogenous state can be modeled by a Markov process. Note that even if the exogenous state process has finite memory, it can be reduced as an equivalent Markov process on a higher-dimensional space. Thus, the Markov assumption is unrestrictive for most practical purposes. By Bellman's principle of optimality, a decision maker aiming to minimize the expected cumulative costs solves the dynamic program

$$
\begin{aligned}
V_t(\boldsymbol{s}_t, \boldsymbol{\xi}_t) = \min_{\boldsymbol{u}_t \in \mathcal{U}_t} \quad & c_t(\boldsymbol{s}_t, \boldsymbol{\xi}_t, \boldsymbol{u}_t) + \mathbb{E}[V_{t+1}(\boldsymbol{s}_{t+1}, \boldsymbol{\xi}_{t+1})|\boldsymbol{\xi}_t] \\
\text{s.t.} \quad & \boldsymbol{s}_{t+1} = g_t(\boldsymbol{s}_t, \boldsymbol{u}_t, \boldsymbol{\xi}_{t+1})
\end{aligned}
\tag{1}
$$

backwards for $t = T, \ldots, 1$ with $V_{T+1} \equiv 0$; see e.g. [1]. The cost-to-go function $V_t(\boldsymbol{s}_t, \boldsymbol{\xi}_t)$ quantifies the minimum expected future cost achievable from state $(\boldsymbol{s}_t, \boldsymbol{\xi}_t)$ at time $t$.

Stochastic optimal control has numerous applications in engineering and science, e.g. in supply chain management, power systems scheduling, behavioral neuroscience, asset allocation, emergency service provisioning, etc. [1, 2]. There is often a natural distinction between endogenous and exogenous states. For example, in inventory control the inventory level can naturally be interpreted as the endogenous state, while the uncertain demand represents the exogenous state.

In spite of their exceptional modeling power, dynamic programming problems of the above type suffer from two major shortcomings that limit their practical applicability. First, the backward induction step (1) is computationally burdensome due to the intractability to evaluate the cost-to-go function $V_t$ for the continuum of all states $(s_t, \boldsymbol{\xi}_t)$, the intractability to evaluate the multivariate conditional expectations and the intractability to optimize over the continuum of all control actions $\boldsymbol{u}_t$ [2]. Secondly, even if the dynamic programming recursions (1) could be computed efficiently, there is often substantial uncertainty about the conditional distribution of $\boldsymbol{\xi}_{t+1}$ given $\boldsymbol{\xi}_t$. Indeed, the distribution of the exogenous states is typically unknown and must be inferred from historical observations. If training data is sparse—as is often the case in practice—it is impossible to estimate this distribution reliably. Thus, we lack essential information to evaluate (1) in the first place.

In this paper, we assume that only a set of $N$ sample trajectories of the exogenous state is given, and we use kernel regression in conjunction with parametric value function approximations to estimate the conditional expectation in (1). Thus, we approximate the conditional distribution of $\boldsymbol{\xi}_{t+1}$ given $\boldsymbol{\xi}_t$ by a discrete distribution whose discretization points are given by the historical samples, while the corresponding conditional probabilities are expressed in terms of a normalized Nadaraya-Watson (NW) kernel function. This data-driven dynamic programming (DDP) approach is conceptually appealing and avoids an artificial separation of estimation and optimization steps. Instead, the historical samples are used directly in the dynamic programming recursions. It is also asymptotically consistent in the sense that the true conditional expectation is recovered when $N$ grows [3]. Moreover, DDP computes the value functions only on the $N$ sample trajectories of the exogenous state, thereby mitigating one of the intractabilities of classical dynamic programming.

Although conceptually and computationally appealing, DDP-based policies exhibit a poor performance in out-of-sample tests if the training data is sparse. In this case the estimate of the conditional expectation in (1) is highly noisy (but largely unbiased). The estimate of the corresponding cost-to-go value inherits this variability. However, it also displays a downward bias caused by the minimization over $\boldsymbol{u}_t$. This phenomenon is reminiscent of overfitting effects in statistics. As estimation errors in the cost-to-go functions are propagated through the dynamic programming recursions, the bias grows over time and thus incentivizes poor control decisions in the early time periods.

The detrimental overfitting effects observed in DDP originate from ignoring distributional uncertainty: DDP takes the estimated discrete conditional distribution of $\boldsymbol{\xi}_{t+1}$ at face value and ignores the possibility of estimation errors. In this paper we propose a robust data-driven dynamic programming (RDDP) approach that replaces the expectation in (1) by a worst-case expectation over a set of distributions close to the nominal estimate in view of the $\chi^2$-distance. We will demonstrate that this regularization reduces both the variability and the bias in the approximate cost-to-go functions and that RDDP dominates ordinary DDP as well as other popular benchmark algorithms in out-of-sample tests. Leveraging on recent results in robust optimization [4] and value function approximation [5] we will also show that the nested min-max problems arising in RDDP typically reduce to conic optimization problems that admit efficient solution with interior point algorithms.

Robust value iteration methods have recently been studied in robust Markov decision process (MDP) theory [6, 7, 8, 9]. However, these algorithms are not fundamentally data-driven as their primitives are uncertainty sets for the transition kernels instead of historical observations. Moreover, they assume finite state and action spaces. Data-driven approaches to dynamic decision making are routinely studied in approximate dynamic programming and reinforcement learning [10, 11, 12], but these methods are not robust (in a worst-case sense) with respect to distributional uncertainty and could therefore be susceptible to overfitting effects. The robust value iterations in RDDP are facilitated by combining convex parametric function approximation methods (to model the dependence on the endogenous state) with nonparametric kernel regression techniques (for the dependence on the exogenous state). This is in contrast to most existing methods, which either rely exclusively on parametric function approximations [10, 11, 13] or nonparametric ones [12, 14, 15, 16]. Due to the convexity in the endogenous state, RDDP further benefits from mathematical programming techniques to optimize over high-dimensional continuous action spaces without requiring any form of discretization.

**Notation.** We use lower-case bold face letters to denote vectors and upper-case bold face letters to denote matrices. We define $\mathbf{1} \in \mathbb{R}^n$ as the vector with all elements equal to 1, while $\Delta = \{\boldsymbol{p} \in \mathbb{R}_+^n : \mathbf{1}^\intercal \boldsymbol{p} = 1\}$ denotes the probability simplex in $\mathbb{R}^n$. The dimensions of $\mathbf{1}$ and $\Delta$ will usually be clear from the context. The space of symmetric matrices of dimension $n$ is denoted by $\mathbb{S}^n$. For any two matrices $\mathbf{X}, \mathbf{Y} \in \mathbb{S}^n$, the relation $\mathbf{X} \succcurlyeq \mathbf{Y}$ implies that $\mathbf{X} - \mathbf{Y}$ is positive semidefinite.

## 2 Data-driven dynamic programming

Assume from now on that the distribution of the exogenous states is unknown and that we are only given $N$ observation histories $\{\boldsymbol{\xi}_t^i\}_{t=1}^T$ for $i = 1, \ldots, N$. This assumption is typically well justified in practice. In this setting, the conditional expectation in (1) cannot be evaluated exactly. However it can be estimated, for instance, via Nadaraya-Watson (NW) kernel regression [17, 18].

$$\mathbb{E}[V_{t+1}(\boldsymbol{s}_{t+1}, \boldsymbol{\xi}_{t+1})|\boldsymbol{\xi}_t] \approx \sum_{i=1}^N q_{ti}(\boldsymbol{\xi}_t)V_{t+1}(\boldsymbol{s}_{t+1}^i, \boldsymbol{\xi}_{t+1}^i) \tag{2}$$

The conditional probabilities in (2) are set to

$$q_{ti}(\boldsymbol{\xi}_t) = \frac{K_{\mathbf{H}}(\boldsymbol{\xi}_t - \boldsymbol{\xi}_t^i)}{\sum_{k=1}^N K_{\mathbf{H}}(\boldsymbol{\xi}_t - \boldsymbol{\xi}_t^k)}, \tag{3}$$

where the kernel function $K_{\mathbf{H}}(\boldsymbol{\xi}) = |\mathbf{H}|^{-\frac{1}{2}} K(|\mathbf{H}|^{-\frac{1}{2}}\boldsymbol{\xi})$ is defined in terms of a symmetric multivariate density $K$ and a positive definite bandwidth matrix $\mathbf{H}$. For a large bandwidth, the conditional probabilities $q_{ti}(\boldsymbol{\xi}_t)$ converge to $\frac{1}{N}$, in which case (2) reduces to the (unconditional) sample average. Conversely, an extremely small bandwidth causes most of the probability mass to be assigned to the sample point closest to $\boldsymbol{\xi}_t$. In the following we set the bandwidth matrix $\mathbf{H}$ to its best estimate assuming that the historical observations $\{\boldsymbol{\xi}_t^i\}_{i=1}^N$ follow a Gaussian distribution; see [19]. Substituting (2) into (1), results in the data-driven dynamic programming (DDP) formulation

$$V_t^{\mathrm{d}}(\boldsymbol{s}_t, \boldsymbol{\xi}_t) = \min_{\boldsymbol{u}_t \in \mathcal{U}_t} \quad c_t(\boldsymbol{s}_t, \boldsymbol{\xi}_t, \boldsymbol{u}_t) + \sum_{i=1}^N q_{ti}(\boldsymbol{\xi}_t) V_{t+1}^{\mathrm{d}}(\boldsymbol{s}_{t+1}^i, \boldsymbol{\xi}_{t+1}^i)$$
$$\text{s.t.} \quad \boldsymbol{s}_{t+1}^i = g_t(\boldsymbol{s}_t, \boldsymbol{u}_t, \boldsymbol{\xi}_{t+1}^i) \quad \forall i, \tag{4}$$

with terminal condition $V_{T+1}^{\mathrm{d}} \equiv 0$. The idea to use kernel-based approximations to estimate the expected future costs is appealing due to its simplicity. Such approximations have been studied, for example, in the context of stochastic optimization with state observation [20]. However, to the best of our knowledge they have not yet been used in a fully dynamic setting—maybe for the reasons to be outlined in § 3. On the positive side, DDP with NW kernel regression is asymptotically consistent for large $N$ under a suitable scaling of the bandwidth matrix and under a mild boundedness assumption on $V_{t+1}^{\mathrm{d}}$ [3]. Moreover, DDP evaluates the cost-to-go function of the next period only at the sample points and thus requires no a-priori discretization of the exogenous state space, thus mitigating one of the intractabilities of classical dynamic programming.

## 3 Robust data-driven dynamic programming

If the training data is sparse, the NW estimate (2) of the conditional expectation in (4) typically exhibits a small bias and a high variability. Indeed, the variance of the estimator scales with $\sim \mathcal{O}(\frac{1}{N})$ [21]. The DDP value function $V_t^{\mathrm{d}}$ inherits this variability. However, it also displays a significant optimistic bias. The following stylized example illustrates this phenomenon.

**Example 3.1** *Assume that $d_1 = 1$, $d_2 = m = 5$, $c_t(\boldsymbol{s}_t, \boldsymbol{\xi}_t, \boldsymbol{u}_t) = 0$, $g_t(\boldsymbol{s}_t, \boldsymbol{u}_t, \boldsymbol{\xi}_{t+1}) = \boldsymbol{\xi}_{t+1}^\mathsf{T} \boldsymbol{u}_t$, $\mathcal{U}_t = \{\boldsymbol{u} \in \mathbb{R}^m : \mathbf{1}^\mathsf{T}\boldsymbol{u} = 1\}$ and $V_{t+1}(s_{t+1}, \boldsymbol{\xi}_{t+1}) = \frac{1}{10}s_{t+1}^2 - s_{t+1}$. In order to facilitate a controlled experiment, we also assume that $(\boldsymbol{\xi}_t, \boldsymbol{\xi}_{t+1})$ follows a multivariate Gaussian distribution, where each component has unit mean and variance. The correlation between $\xi_{t,k}$ and $\xi_{t+1,k}$ is set to 30%. All other correlations are zero. Our aim is to solve (1) and to estimate $V_t(s_t, \boldsymbol{\xi}_t)$ at $\boldsymbol{\xi}_t = \mathbf{1}$.*

*By permutation symmetry, the optimal decision under full distributional knowledge is $\boldsymbol{u}_t^* = \frac{1}{5}\mathbf{1}$. An analytical calculation then yields the true cost-to-go value $V_t(s_t, \mathbf{1}) = -0.88$. In the following we completely ignore our distributional knowledge. Instead, we assume that only $N$ independent samples $(\boldsymbol{\xi}_t^i, \boldsymbol{\xi}_{t+1}^i)$ are given, $i = 1, \ldots, N$. To showcase the high variability of NW estimation, we fix the decision $\boldsymbol{u}_t^*$ and use (2) to estimate its expected cost conditional on $\boldsymbol{\xi}_t = \mathbf{1}$. Figure 1 (left) shows that this estimator is unbiased but fluctuates within $\pm 5\%$ around its median even for $N = 500$. Next, we use (4) to estimate $V_t^{\mathrm{d}}(s_t, \mathbf{1})$, that is, the expected cost of the best decision obtained without distributional information. Figure 1 (middle) shows that this cost estimator is even more noisy than the one for a fixed decision, exhibits a significant downward bias and converges slowly as $N$ grows.*

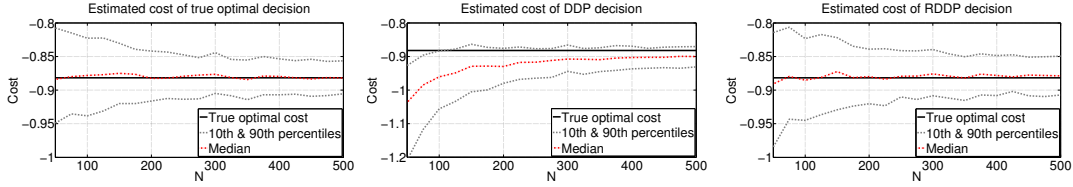

Figure 1: Estimated costs of true optimal and data-driven decisions. Note the different scales. All reported values represent averages over 200 independent simulation runs.

The downward bias in $V_t^{\mathrm{d}}$ as an estimator for the true value function $V_t$ is the consequence of an overfitting effect, which can be explained as follows. Setting $V_{t+1} \equiv V_{t+1}^{\mathrm{d}}$, we find

$$V_t(\boldsymbol{s}_t, \boldsymbol{\xi}_t) = \min_{\boldsymbol{u}_t \in \mathcal{U}_t} c_t(\boldsymbol{s}_t, \boldsymbol{\xi}_t, \boldsymbol{u}_t) + \mathbb{E}[V_{t+1}^{\mathrm{d}}(g_t(\boldsymbol{s}_t, \boldsymbol{u}_t, \boldsymbol{\xi}_{t+1}), \boldsymbol{\xi}_{t+1})|\boldsymbol{\xi}_t]$$

$$\approx \min_{\boldsymbol{u}_t \in \mathcal{U}_t} c_t(\boldsymbol{s}_t, \boldsymbol{\xi}_t, \boldsymbol{u}_t) + \mathbb{E}[\sum_{i=1}^{N} q_{ti}(\boldsymbol{\xi}_t) V_{t+1}^{\mathrm{d}}(g_t(\boldsymbol{s}_t, \boldsymbol{u}_t, \boldsymbol{\xi}_{t+1}^i), \boldsymbol{\xi}_{t+1}^i)|\boldsymbol{\xi}_t]$$

$$\geq \mathbb{E}\Big[\min_{\boldsymbol{u}_t \in \mathcal{U}_t} c_t(\boldsymbol{s}_t, \boldsymbol{\xi}_t, \boldsymbol{u}_t) + \sum_{i=1}^{N} q_{ti}(\boldsymbol{\xi}_t) V_{t+1}^{\mathrm{d}}(g_t(\boldsymbol{s}_t, \boldsymbol{u}_t, \boldsymbol{\xi}_{t+1}^i), \boldsymbol{\xi}_{t+1}^i)\Big|\boldsymbol{\xi}_t\Big].$$

The relation in the second line uses our observation that the NW estimator of the expected cost associated with any fixed decision $\boldsymbol{u}_t$ is approximately unbiased. Here, the expectation is with respect to the (independent and identically distributed) sample trajectories used in the NW estimator. The last line follows from the conditional Jensen inequality. Note that the expression inside the conditional expectation coincides with $V_t^{\mathrm{d}}(\boldsymbol{s}_t, \boldsymbol{\xi}_t)$. This argument suggests that $V_t^{\mathrm{d}}(\boldsymbol{s}_t, \boldsymbol{\xi}_t)$ must indeed underestimate $V_t(\boldsymbol{s}_t, \boldsymbol{\xi}_t)$ on average. We emphasize that all systematic estimation errors of this type accumulate as they are propagated through the dynamic programming recursions.

To mitigate the detrimental overfitting effects, we propose a regularization that reduces the decision maker's overconfidence in the weights $\boldsymbol{q}_t(\boldsymbol{\xi}_t) = [q_{t1}(\boldsymbol{\xi}_t) \ldots q_{tN}(\boldsymbol{\xi}_t)]^\mathsf{T}$. Thus, we allow the conditional probabilities used in (4) to deviate from their nominal values $\boldsymbol{q}_t(\boldsymbol{\xi}_t)$ up to a certain degree. This is achieved by considering uncertainty sets $\Delta(\boldsymbol{q})$ that contain all weight vectors sufficiently close to some nominal weight vector $\boldsymbol{q} \in \Delta$ with respect to the $\chi^2$-distance for histograms.

$$\Delta(\boldsymbol{q}) = \{\boldsymbol{p} \in \Delta : \sum_{i=1}^{N} (p_i - q_i)^2 / p_i \leq \gamma\} \tag{5}$$

The $\chi^2$-distance belongs to the class of $\phi$-divergences [22], which also includes the Kullback-Leibler distances. Our motivation for using uncertainty sets of the type (5) is threefold. First, $\Delta(\boldsymbol{q})$ is determined by a single size parameter $\gamma$, which can easily be calibrated, e.g., via cross-validation. Secondly, the $\chi^2$-distance guarantees that any distribution $\boldsymbol{p} \in \Delta(\boldsymbol{q})$ assigns nonzero probability to all scenarios that have nonzero probability under the nominal distribution $\boldsymbol{q}$. Finally, the structure of $\Delta(\boldsymbol{q})$ implied by the $\chi^2$-distance has distinct computational benefits that become evident in § 4.

Allowing the conditional probabilities in (4) to range over the uncertainty set $\Delta(\boldsymbol{q}_t(\boldsymbol{\xi}_t))$ results in the robust data-driven dynamic programming (RDDP) formulation

$$V_t^{\mathrm{r}}(\boldsymbol{s}_t, \boldsymbol{\xi}_t) = \min_{\boldsymbol{u}_t \in \mathcal{U}_t} \quad c_t(\boldsymbol{s}_t, \boldsymbol{\xi}_t, \boldsymbol{u}_t) + \max_{\boldsymbol{p} \in \Delta(\boldsymbol{q}_t(\boldsymbol{\xi}_t))} \sum_{i=1}^{N} p_i V_{t+1}^{\mathrm{r}}(\boldsymbol{s}_{t+1}^i, \boldsymbol{\xi}_{t+1}^i)$$

$$\text{s.t.} \quad \boldsymbol{s}_{t+1}^i = g_t(\boldsymbol{s}_t, \boldsymbol{u}_t, \boldsymbol{\xi}_{t+1}^i) \quad \forall i \tag{6}$$

with terminal condition $V_{T+1}^{\mathrm{r}} \equiv 0$. Thus, each RDDP recursion involves the solution of a robust optimization problem [4], which can be viewed as a game against 'nature' (or a malicious adversary): for every action $\boldsymbol{u}_t$ chosen by the decision maker, nature selects the corresponding worst-case weight vector from within $\boldsymbol{p} \in \Delta(\boldsymbol{q}_t(\boldsymbol{\xi}_t))$. By anticipating nature's moves, the decision maker is forced to select more conservative decisions that are less susceptible to amplifying estimation errors in the nominal weights $\boldsymbol{q}_t(\boldsymbol{\xi}_t)$. The level of robustness of the RDDP scheme can be steered by selecting

the parameter $\gamma$. We suggest to choose $\gamma$ large enough such that the envelope of all conditional CDFs of $\boldsymbol{\xi}_{t+1}$ implied by the weight vectors in $\Delta(\boldsymbol{q}_t(\boldsymbol{\xi}_t))$ covers the true conditional CDF with high confidence (Figure 2). The following example illustrates the potential benefits of the RDDP scheme.

**Example 3.2** *Consider again Example 3.1. Assuming that only the samples $\{\boldsymbol{\xi}_t^i, \boldsymbol{\xi}_{t+1}^i\}_{i=1}^N$ are known, we can compute a worst-case optimal decision using (6). Fixing this decision, we can then use (2) to estimate its expected cost conditional on $\boldsymbol{\xi}_t = \mathbf{1}$. Note that this cost is generically different from $V_t^r(s_t, \mathbf{1})$. Figure 1 (right) shows that the resulting cost estimator is less noisy and—perhaps surprisingly—unbiased. Thus, it clearly dominates $V_t^d(s_t, \mathbf{1})$ as an estimator for the true cost-to-go value $V_t(s_t, \mathbf{1})$ (which is not accessible in reality as it relies on full distributional information).*

Robust optimization models with uncertainty sets of the type (5) have previously been studied in [23, 24]. However, these static models are fundamentally different in scope from our RDDP formulation. RDDP seeks the worst-case probabilities of $N$ historical samples of the exogenous state, using the NW weights as nominal probabilities. In contrast, the static models in [23, 24] rely on a partition of the uncertainty space into $N$ bins. Worst-case probabilities are then assigned to the bins, whose nominal probabilities are given by the empirical frequencies. This latter approach does not seem to extend easily to our dynamic setting as it would be unclear where in each bin one should evaluate the cost-to-go functions.

Instead of immunizing the DDP scheme against estimation errors in the conditional probabilities (as advocated here), one could envisage other regularizations to mitigate the overfitting phenomena. For instance, one could construct an uncertainty set for $(\boldsymbol{\xi}_{t+1}^i)_{i=1}^N$ and seek control actions that are optimal in view of the worst-case sample points within this set. However, this approach would lead to a harder robust optimization problem, where the search space of the inner maximization has dimension $\mathcal{O}(Nd_2)$ (as opposed to $\mathcal{O}(N)$ for RDDP). Moreover, this approach would only be tractable if $V_{t+1}^r$ displayed a very regular (e.g., linear or quadratic) dependence on $\boldsymbol{\xi}_{t+1}$. RDDP imposes no such restrictions on the cost-to-go function; see § 4.

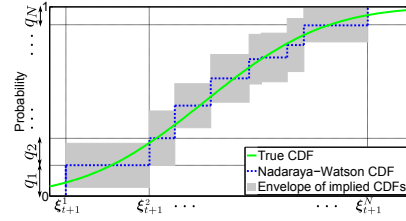

Figure 2: Envelope of all conditional CDFs implied by weight vectors in $\Delta(\boldsymbol{q}_t(\boldsymbol{\xi}_t))$.

## 4 Computational solution procedure

In this section we demonstrate that RDDP is computationally tractable under a convexity assumption and if we approximate the dependence of the cost-to-go functions on the endogenous state through a piecewise linear or quadratic approximation architecture. This result immediately extends to the DDP scheme of § 2 as the uncertainty set (5) collapses to a singleton for $\gamma = 0$.

**Assumption 4.1** *For all $t = 1, \ldots, T$, the cost function $c_t$ is convex quadratic in $(s_t, \boldsymbol{u}_t)$, the transition function $g_t$ is affine in $(s_t, \boldsymbol{u}_t)$, and the feasible set $\mathcal{U}_t$ is second-order conic representable.*

Under Assumption 4.1, $V_t^r(s_t, \boldsymbol{\xi}_t)$ can be evaluated by solving a convex optimization problem.

**Theorem 4.1** *Suppose that Assumption 4.1 holds and that the cost-to-go function $V_{t+1}^r$ is convex in the endogenous state. Then, (6) reduces to the following convex minimization problem.*

$$
\begin{aligned}
V_t^r(s_t, \boldsymbol{\xi}_t) = \min \quad & c_t(s_t, \boldsymbol{\xi}_t, \boldsymbol{u}_t) + \lambda\gamma - \mu - 2\boldsymbol{q}_t(\boldsymbol{\xi}_t)^\mathsf{T}\boldsymbol{y} + 2\lambda\boldsymbol{q}_t(\boldsymbol{\xi}_t)^\mathsf{T}\mathbf{1} \\
\text{s.t.} \quad & \boldsymbol{u}_t \in \mathcal{U}_t, \quad \mu \in \mathbb{R}, \quad \lambda \in \mathbb{R}_+, \quad \boldsymbol{z}, \boldsymbol{y} \in \mathbb{R}^N \\
& V_{t+1}^r(g_t(s_t, \boldsymbol{u}_t, \boldsymbol{\xi}_{t+1}^i), \boldsymbol{\xi}_{t+1}^i) \le z_i \quad \forall i \\
& z_i + \mu \le \lambda, \quad \sqrt{4y_i^2 + (z_i + \mu)^2} \le 2\lambda - z_i - \mu \quad \forall i
\end{aligned}
\tag{7}
$$

**Corollary 4.1** *If Assumption 4.1 holds, then RDDP preserves convexity in the exogenous state. Thus, $V_t^r(s_t, \boldsymbol{\xi}_t)$ is convex in $s_t$ whenever $V_{t+1}^r(s_{t+1}, \boldsymbol{\xi}_{t+1})$ is convex in $s_{t+1}$.*

Note that problem (7) becomes a tractable second-order cone program if $V_{t+1}^r$ is convex piecewise linear or convex quadratic in $s_{t+1}$. Then, it can be solved efficiently with interior point algorithms.

---

<div align="center">Algorithm 1: Robust data-driven dynamic programming</div>

---

**Inputs:** Sample trajectories $\{s_t^k\}_{t=1}^T$ for $k = 1, \ldots, K$;
   observation histories $\{\xi_t^i\}_{t=1}^{T+1}$ for $i = 1, \ldots, N$.
**Initialization:** Let $\hat{V}_{T+1}^{\mathrm{r}}(\cdot, \xi_{T+1}^i)$ be the zero function for all $i = 1, \ldots, N$.
**for all** $t = T, \ldots, 1$ **do**
   **for all** $i = 1, \ldots, N$ **do**
      **for all** $k = 1, \ldots, K$ **do**
         Let $\hat{V}_{t,k,i}^{\mathrm{r}}$ be the optimal value of problem (7) with input $\hat{V}_{t+1}^{\mathrm{r}}(\cdot, \xi_{t+1}^j)$ $\forall j$.
      **end for**
      Construct $\hat{V}_t^{\mathrm{r}}(\cdot, \xi_t^i)$ from the interpolation points $\{(s_t^k, \hat{V}_{t,k,i}^{\mathrm{r}})\}_{k=1}^K$ as in (8a) or (8b).
   **end for**
**end for**
**Outputs:** Approximate cost-to-go functions $\hat{V}_t^{\mathrm{r}}(\cdot, \xi_t^i)$ for $i = 1, \ldots, N$ and $t = 1, \ldots, T$.

---

We now describe an algorithm that computes all cost-to-go functions $\{V_t^{\mathrm{r}}\}_{t=1}^T$ approximately. Initially, we collect historical observation trajectories of the exogenous state $\{\xi_t^i\}_{t=1}^T$, $i = 1, \ldots, N$, and generate sample trajectories of the endogenous state $\{s_t^k\}_{t=1}^T$, $k = 1, \ldots, K$, by simulating the evolution of $s_t$ under a prescribed control policy along randomly selected exogenous state trajectories. Best results are achieved if the sample-generating policy is near-optimal. If no near-optimal policy is known, an initial naive policy can be improved sequentially in a greedy fashion. The core of the algorithm computes approximate value functions $\hat{V}_t^{\mathrm{r}}$, which are piecewise linear or quadratic in $s_t$, by backward induction on $t$. Iteration $t$ takes $\hat{V}_{t+1}^{\mathrm{r}}$ as an input and computes the optimal value $\hat{V}_{t,k,i}^{\mathrm{r}}$ of the second-order cone program (7) for each sample state $(s_t^k, \xi_t^i)$. For any fixed $i$ we then construct the function $\hat{V}_t^{\mathrm{r}}(\cdot, \xi_t^i)$ from the interpolation points $\{(s_t^k, \hat{V}_{t,k,i}^{\mathrm{r}})\}_{k=1}^K$. If the endogenous state is univariate ($d_1 = 1$), the following piecewise linear approximation is used.

$$\hat{V}_t^{\mathrm{r}}(s_t, \xi_t^i) = \max_k (s_t^k - s_t)/(s_t^k - s_t^{k-1})\hat{V}_{t,k-1,i}^{\mathrm{r}} + (s_t - s_t^{k-1})/(s_t^k - s_t^{k-1})\hat{V}_{t,k,i}^{\mathrm{r}} \tag{8a}$$

In the multivariate case ($d_1 > 1$), we aim to find the convex quadratic function $\hat{V}_t^{\mathrm{r}}(s_t, \xi_t^i) = s_t^\mathsf{T} M_i s_t + 2 m_i^\mathsf{T} s_t + m_i$ that best explains the given interpolation points in a least-squares sense. This quadratic function can be computed efficiently by solving the following semidefinite program.

$$
\begin{aligned}
\min \quad & \sum_{k=1}^K \left[ (s_t^k)^\mathsf{T} M_i s_t^k + 2 m_i^\mathsf{T} s_t^k + m_i - \hat{V}_{t,k,i}^{\mathrm{r}} \right]^2 \\
\text{s.t.} \quad & M_i \in \mathbb{S}^{d_1}, \quad M_i \succcurlyeq 0, \quad m_i \in \mathbb{R}^{d_1}, \quad m_i \in \mathbb{R}
\end{aligned}
\tag{8b}
$$

Quadratic approximation architectures of the above type first emerged in approximate dynamic programming [5]. Once the function $\hat{V}_t^{\mathrm{r}}(\cdot, \xi_t^i)$ is computed for all $i = 1, \ldots, N$, the algorithm proceeds to iteration $t - 1$. A summary of the overall procedure is provided in Algorithm 1.

**Remark 4.1** *The RDDP algorithm remains valid if the feasible set $\mathcal{U}_t$ depends on the state $(s_t, \xi_t)$ or if the control action $u_t$ includes components that are of the 'here-and-now'-type (i.e., they are chosen before $\xi_{t+1}$ is observed) as well as others that are of the 'wait-and-see'-type (i.e., they are chosen after $\xi_{t+1}$ has been revealed). In this setting, problem (7) becomes a two-stage stochastic program [25] but remains efficiently solvable as a second-order cone program.*

## 5 Experimental results

We evaluate the RDDP algorithm of § 4 in the context of an index tracking and a wind energy commitment application. All semidefinite programs are solved with SeDuMi [26] by using the Yalmip [27] interface, while all linear and second-order cone programs are solved with CPLEX.

### 5.1 Index tracking

The objective of index tracking is to match the performance of a stock index as closely as possible with a portfolio of other financial instruments. In our experiment, we aim to track the S&P 500

| Statistic | LSPI | DDP | RDDP |
|-----------|------|-----|------|
| Mean | 5.692 | 4.697 | 1.285 |
| Std. dev. | 11.699 | 15.067 | 2.235 |
| 90th prct. | 14.597 | 9.048 | 2.851 |
| Worst case | 126.712 | 157.201 | 18.832 |

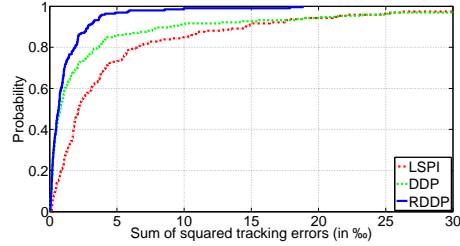

Table 1: Out-of-sample statistics of sum of squared tracking errors in ‰.

Figure 3: Cumulative distribution function of sum of squared tracking errors.

index with a combination of the NASDAQ Composite, Russell 2000, S&P MidCap 400, and AMEX Major Market indices. We set the planning horizon to $T = 20$ trading days (1 month).

Let $s_t \in \mathbb{R}_+$ be the value of the current tracking portfolio relative to the value of S&P 500 on day $t$, while $\boldsymbol{\xi}_t \in \mathbb{R}_+^5$ denotes the vector of the total index returns (price relatives) from day $t-1$ to day $t$. The first component of $\boldsymbol{\xi}_t$ represents the return of S&P 500. The objective of index tracking is to maintain $s_t$ close to 1 in a least-squares sense throughout the planning horizon, which gives rise to the following dynamic program with terminal condition $V_{T+1} \equiv 0$.

$$
\begin{aligned}
V_t(s_t, \boldsymbol{\xi}_t) = \quad &\min \quad (1 - s_t)^2 + \mathbb{E}[V_t(s_{t+1}, \boldsymbol{\xi}_{t+1}) | \boldsymbol{\xi}_t] \\
&\text{s.t.} \quad \boldsymbol{u} \in \mathbb{R}_+^5, \quad \mathbf{1}^\intercal \boldsymbol{u} = s_t, \quad u_1 = 0, \quad s_{t+1} = \boldsymbol{\xi}_{t+1}^\intercal \boldsymbol{u} / \xi_{t+1,1}
\end{aligned}
\tag{9}
$$

Here, $u_i / s_t$ can be interpreted as the portion of the tracking portfolio that is invested in index $i$ on day $t$. Our computational experiment is based on historical returns of the indices over 5440 days from 26-Aug-1991 to 8-Mar-2013 (272 trading months). We solve the index tracking problem using the DDP and RDDP algorithms (i.e., the algorithm of § 4 with $\gamma = 0$ and $\gamma = 10$, respectively) as well as least-squares policy iteration (LSPI) [10]. As the endogenous state is univariate, DDP and RDDP employ the piecewise linear approximation architecture (8a). LSPI solves an infinite-horizon variant of problem (9) with discount factor $\lambda = 0.9$, polynomial basis features of degree 3 and a discrete action space comprising 1,000 points sampled uniformly from the true continuous action space. We train the algorithms on the first 80 and test on the remaining 192 trading months.

Table 1 reports several out-of-sample statistics of the sum of squared tracking errors. We find that RDDP outperforms DDP and LSPI by a factor of 4-5 in view of the mean, the standard deviation and the 90th percentile of the error distribution, and it outperforms the other algorithms by an order of magnitude in view of the worst-case (maximum) error. Figure 3 further shows that the error distribution generated by RDDP stochastically dominates those generated by DDP and LSPI.

## 5.2 Wind energy commitment

Next, we apply RDDP to the wind energy commitment problem proposed in [28, 29]. On every day $t$, a wind energy producer chooses the energy commitment levels $\boldsymbol{x}_t \in \mathbb{R}_+^{24}$ for the next 24

| Site | Statistic | Persistence | DDP | RDDP |
|------|-----------|-------------|-----|------|
| NC | Mean | 4.039 | 4.698 | 7.549 |
| | Std. dev. | 3.964 | 6.338 | 5.133 |
| | 10th prct. | 0.524 | -1.463 | 1.809 |
| | Worst case | -11.221 | -22.666 | 0.481 |
| OH | Mean | 2.746 | 4.104 | 5.510 |
| | Std. dev. | 3.428 | 5.548 | 4.500 |
| | 10th prct. | 0.154 | 0.118 | 1.395 |
| | Worst case | -12.065 | -21.317 | 0.280 |

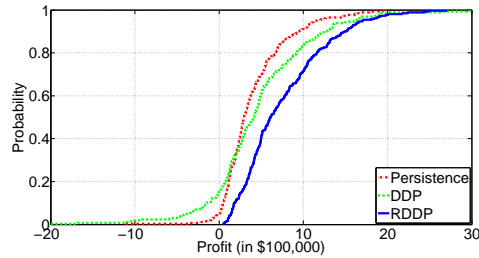

Table 2: Out-of-sample statistics of profit (in $100,000).

Figure 4: Out-of-sample profit distribution for the North Carolina site.

hours. The day-ahead prices $\boldsymbol{\pi}_t \in \mathbb{R}_+^{24}$ per unit of energy committed are known at the beginning of the day. However, the hourly amounts of wind energy $\boldsymbol{\omega}_{t+1} \in \mathbb{R}_+^{24}$ generated over the day are uncertain. If the actual production falls short of the commitment levels, there is a penalty of twice the respective day-ahead price for each unit of unsatisfied demand. The wind energy producer also operates three storage devices indexed by $l \in \{1, 2, 3\}$, each of which can have a different capacity $\overline{s}^l$, hourly leakage $\rho^l$, charging efficiency $\rho_c^l$ and discharging efficiency $\rho_d^l$. We denote by $\boldsymbol{s}_{t+1}^l \in \mathbb{R}_+^{24}$ the hourly filling levels of storage $l$ over the next 24 hours. The wind producer's objective is to maximize the expected profit over a short-term planning horizon of $T = 7$ days.

The endogenous state is given by the storage levels at the end of day $t$, $\boldsymbol{s}_t = \{s_{t,24}^l\}_{l=1}^3 \in \mathbb{R}_+^3$, while the exogenous state comprises the day-ahead prices $\boldsymbol{\pi}_t \in \mathbb{R}_+^{24}$ and the wind energy production levels $\boldsymbol{\omega}_t \in \mathbb{R}_+^{24}$ of day $t-1$, which are revealed to the producer on day $t$. Thus, we set $\boldsymbol{\xi}_t = (\boldsymbol{\pi}_t, \boldsymbol{\omega}_t)$. The best bidding and storage strategy can be found by solving the dynamic program

$$
\begin{aligned}
V_t(\boldsymbol{s}_t, \boldsymbol{\xi}_t) = \quad \max \quad & \boldsymbol{\pi}_t^\mathsf{T} \boldsymbol{x}_t - 2\boldsymbol{\pi}_t^\mathsf{T} \mathbb{E}[\boldsymbol{e}_{t+1}^u | \boldsymbol{\xi}_t] + \mathbb{E}[V_{t+1}(\boldsymbol{s}_{t+1}, \boldsymbol{\xi}_{t+1}) | \boldsymbol{\xi}_t] \\
\text{s.\,t.} \quad & \boldsymbol{x}_t, \boldsymbol{e}_{t+1}^{\{c,w,u\}} \in \mathbb{R}_+^{24}, \quad \boldsymbol{e}_{t+1}^{\{+,-\},l}, \boldsymbol{s}_{t+1}^l \in \mathbb{R}_+^{24} \quad \forall l \\
& \omega_{t+1,h} = e_{t+1,h}^c + e_{t+1,h}^{+,1} + e_{t+1,h}^{+,2} + e_{t+1,h}^{+,3} + e_{t+1,h}^w \quad \forall h \\
& x_{t,h} = e_{t+1,h}^c + e_{t+1,h}^{-,1} + e_{t+1,h}^{-,2} + e_{t+1,h}^{-,3} + e_{t+1,h}^u \quad \forall h \\
& s_{t+1,h}^l = \rho^l s_{t+1,h-1}^l + \rho_c^l e_{t+1,h}^{+,l} - \frac{1}{\rho_d^l} e_{t+1,h}^{-,l}, \quad s_{t+1,h}^l \leq \overline{s}^l \quad \forall h, l
\end{aligned}
\tag{10}
$$

with terminal condition $V_{T+1} \equiv 0$. Here, we adopt the convention that $s_{t+1,0}^l = s_{t,24}^l$ for all $l$. Besides the usual here-and-now decisions $\boldsymbol{x}_t$, the decision vector $\boldsymbol{u}_t$ now also includes wait-and-see decisions that are chosen after $\boldsymbol{\xi}_{t+1}$ has been revealed (see Remark 4.1): $\boldsymbol{e}^c$ represents the amount of wind energy used to meet the commitment, $\boldsymbol{e}^{+,l}$ represents the amount of wind energy fed into storage $l$, $\boldsymbol{e}^{-,l}$ represents the amount of energy from storage $l$ used to meet the commitment, $\boldsymbol{e}^w$ represents the amount of wind energy that is wasted, and $\boldsymbol{e}^u$ represents the unmet energy commitment.

Our computational experiment is based on day-ahead prices for the PJM market and wind speed data for North Carolina (33.9375N, 77.9375W) and Ohio (41.8125N, 81.5625W) from 2002 to 2011 (520 weeks). As $\boldsymbol{\xi}_t$ is a 48 dimensional vector with high correlations between its components, we perform principal component analysis to obtain a 6 dimensional subspace that explains more than $90\%$ of the variability of the historical observations. The conditional probabilities $\boldsymbol{q}_t(\boldsymbol{\xi}_t)$ are subsequently estimated using the projected data points. The parameters for the storage devices are taken from [30]. We solve the wind energy commitment problem using the DDP and RDDP algorithms (i.e., the algorithm of § 4 with $\gamma = 0$ and $\gamma = 1$, respectively) as well as a persistence heuristic that naively pledges the wind generation of the previous day by setting $\boldsymbol{x}_t = \boldsymbol{\omega}_t$. Persistence was proposed as a useful baseline in [28]. Note that problem (10) is beyond the scope of traditional reinforcement learning algorithms due to the high dimensionality of the action spaces and the seasonalities in the wind and price data. We train DDP and RDDP on the first 260 weeks and test the resulting commitment strategies as well as the persistence heuristic on the last 260 weeks of the data set.

Table 2 reports the test statistics of the different algorithms. We find that the persistence heuristic wins in terms of standard deviation, while RDDP wins in all other categories. However, the higher standard deviation of RDDP can be explained by a heavier upper tail (which is indeed desirable). Moreover, the profit distribution generated by RDDP stochastically dominates those generated by DDP and the persistence heuristic; see Figure 4. Another major benefit of RDDP is that it cuts off any losses (negative profits), whereas all other algorithms bear a significant risk of incurring a loss.

**Concluding remarks** The proposed RDDP algorithm combines ideas from robust optimization, reinforcement learning and approximate dynamic programming. We remark that the $NK$ convex optimization problems arising in each backward induction step are independent of each other and thus lend themselves to parallel implementation. We also emphasize that Assumption 4.1 could be relaxed to allow $c_t$ and $g_t$ to display a general nonlinear dependence on $\boldsymbol{s}_t$. This would invalidate Corollary 4.1 but not Theorem 4.1. If one is willing to accept a potentially larger mismatch between the true nonconvex cost-to-go function and its convex approximation architecture, then Algorithm 1 can even be applied to specific motor control, vehicle control or other nonlinear control problems.

**Acknowledgments:** This research was supported by EPSRC under grant EP/I014640/1.

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
