[Supplementary Material]

# Supplement to 'Robust Data-Driven Dynamic Programming'

## Abstract

This supplement contains proofs and technical background material omitted from the main text.

## A   Proofs

The following lemma constitutes a key ingredient for the proof of Theorem 4.1.

**Lemma A.1** *Select $\boldsymbol{q} \in \Delta$ and $\boldsymbol{v} \in \mathbb{R}^N$, and define $\Delta(\boldsymbol{q})$ as in (5). Then, the worst-case expectation $\max_{\boldsymbol{p} \in \Delta(\boldsymbol{q})} \boldsymbol{v}^\intercal \boldsymbol{p}$ can be computed by solving the following tractable second-order cone program.*

$$
\begin{aligned}
\min \quad & \lambda\gamma - \mu - 2\boldsymbol{q}^\intercal \boldsymbol{y} + 2\lambda \boldsymbol{q}^\intercal \mathbf{1} \\
\text{s.t.} \quad & \mu \in \mathbb{R}, \quad \lambda \in \mathbb{R}_+, \quad \boldsymbol{z}, \boldsymbol{y} \in \mathbb{R}^N \\
& v_i \leq z_i, \quad z_i + \mu \leq \lambda, \quad \sqrt{4y_i^2 + (z_i + \mu)^2} \leq 2\lambda - z_i - \mu \quad \forall i
\end{aligned}
\tag{A.1}
$$

**Proof:** Using the definition of $\Delta(\boldsymbol{q})$ in (5), we can express the worst-case expectation problem as

$$
\begin{aligned}
\max_{\boldsymbol{p} \in \mathbb{R}_+^N} \quad & \boldsymbol{v}^\intercal \boldsymbol{p} \\
\text{s.t.} \quad & \mathbf{1}^\intercal \boldsymbol{p} = 1, \quad \sum_{i=1}^N \frac{(p_i - q_i)^2}{p_i} \leq \gamma \, .
\end{aligned}
\tag{A.2}
$$

The corresponding Lagrangian is given by

$$
\mathcal{L}(\boldsymbol{p}, \lambda, \mu) \quad = \lambda\gamma - \mu + \sum_{i=1}^N (v_i + \mu)p_i - \lambda \frac{(p_i - q_i)^2}{p_i} \, .
$$

Next, we can use the Lagrangian to dualize (A.2),

$$
\max_{\boldsymbol{p} \in \Delta(\boldsymbol{q})} \boldsymbol{v}^\intercal \boldsymbol{p} \; = \; \max_{\boldsymbol{p} \in \mathbb{R}_+^N} \min_{\mu \in \mathbb{R}, \lambda \in \mathbb{R}_+} \mathcal{L}(\boldsymbol{p}, \lambda, \mu) \; = \; \min_{\mu \in \mathbb{R}, \lambda \in \mathbb{R}_+} \max_{\boldsymbol{p} \in \mathbb{R}_+^N} \mathcal{L}(\boldsymbol{p}, \lambda, \mu),
$$

where the second equality follows from strong duality, which holds as $\boldsymbol{q}$ constitutes a Slater point in the relative interior of the primal problem's feasible set. After some elementary manipulations we find that the optimal value of (A.2) is expressible as

$$
\min_{\mu \in \mathbb{R}, \lambda \in \mathbb{R}_+} \lambda\gamma - \mu + \sum_{i=1}^N \max_{p_i \in \mathbb{R}_+} \left( (v_i + \mu)p_i - \lambda \frac{(p_i - q_i)^2}{p_i} \right) \, .
\tag{A.3}
$$

Consider now the $i$th inner subproblem in (A.3). If $q_i = 0$, then the subproblem reduces to

$$
\max_{p_i \in \mathbb{R}_+} (v_i + \mu - \lambda)p_i = \left\{ \begin{array}{ll} 0 & \text{if } \lambda - v_i - \mu \geq 0, \\ \infty & \text{otherwise.} \end{array} \right.
$$

If $q_i > 0$ and $\lambda - v_i - \mu \geq 0$, then the subproblem has the analytical solution $p_i^* = q_i\sqrt{\lambda/(\lambda - v_i - \mu)}$ with corresponding optimal value $2\lambda q_i - 2q_i\sqrt{\lambda(\lambda - v_i - \mu)}$. On the other hand, if $q_i > 0$ and $\lambda - v_i - \mu < 0$, the problem is unbounded. Substituting the explicit solutions of all subproblems into (A.3) shows that the optimal value of (A.2) is given by

$$\min_{\mu \in \mathbb{R}, \lambda \in \mathbb{R}_+} \quad \lambda\gamma - \mu + 2\lambda \boldsymbol{q}^\mathsf{T}\mathbf{1} - 2\sum_{i=1}^{N} q_i\sqrt{\lambda(\lambda - v_i - \mu)}$$

$$\text{s.t.} \quad \lambda - v_i - \mu \geq 0 \quad \forall i. \tag{A.4}$$

It is now easy to see that problem (A.4) is equivalent to (A.1), and thus the claim follows. ∎

**Proof of Theorem 4.1:** The claim is an immediate consequence of Lemma A.1. Substituting (A.1) into the min-max problem (6) yields the desired result. ∎

**Proof of Corollary 4.1:** The claim follows from [1, Theorem 1], which asserts that the partial minimum (taken only with respect of a subset of all variables) of a convex function is convex. ∎

# B   Specifics of the RDDP algorithm

This section addresses implementational details of the RDDP algorithm. In Section B.1 we outline a procedure for constructing sample trajectories of the endogenous state, and in Section B.2 we describe the selection of the algorithm's design parameters.

## B.1   Generating sample trajectories

The sample trajectories $\{\boldsymbol{s}_t^k\}_{t=1}^T$, $k = 1, \ldots, K$, which are needed as inputs for Algorithm 1, can be obtained by simulating a given policy along randomly selected exogenous state trajectories. Best results are achieved if the sample-generating policy is near-optimal. If no near-optimal policy is known, an initial naive policy can be improved sequentially in a greedy fashion [2]. For constrained linear-quadratic regulator (LQR) problems, we use the exact optimal policy of the corresponding *un*constrained LQR problem as the initial policy. In all other cases, we start with a naive model predictive control policy. The underlying exogenous state trajectories (along which the endogenous state is simulated) are obtained from the historical trajectories $\{\boldsymbol{\xi}_t^i\}_{t=1}^T$, $i = 1, \ldots, N$, by allowing random inter-trajectory crossovers according to the conditional probabilities (3).

If the endogenous state has low dimension (e.g., $d_1 < 4$), the evaluation points $\{\boldsymbol{s}_t^k\}_{t=1}^T$, $k = 1, \ldots, K$, can be sampled uniformly from the set of all feasible endogenous states; see e.g. [3]. Similarly, if the control objective is to track a prescribed target, the evaluation points can be obtained by sampling states in the target's vicinity; see [2, Section 7].

## B.2   Parameter selection

The RDDP algorithm is parameterized by the level of robustness $\gamma$, the bandwidth matrix $\mathbf{H}$ and the number of sample trajectories $K$. We choose $\gamma$ via cross-validation from within the set $\{0.1, 1, 10\}$. Note that $\gamma$ should decrease as the number of observation histories $N$ grows. The matrix $\mathbf{H}$ could also be obtained via cross-validation. However, we set $\mathbf{H} = \text{diag}(h_1^2, \ldots, h_{d_2}^2)$, where

$$h_j = \hat{\sigma}_j \left(\frac{4}{N(d_2 + 2)}\right)^{\frac{1}{d_2+4}} \quad \forall j = 1, \ldots, d_2,$$

and $\hat{\sigma}_j$ denotes the (sample) standard deviation of the $j$-th component of the exogenous state. This choice of $\mathbf{H}$ yields an asymptotically consistent estimator for the exogenous state distribution. Moreover, it minimizes the mean integrated square error if the exogenous state is Gaussian; see [4]. Finally, we choose $K$ large enough to ensure that the approximate value function at the sample points does not change significantly (in terms of the $\ell_2$-norm) when new samples are added. The number of historical observations $N$ can principally be selected in a similar manner as $K$. However, in practice we typically use all the available historical observations as $N$ is assumed to be small.