[Reviews · NeurIPS 2013]

Submitted by Assigned_Reviewer_4

This paper considers finite-horizon stochastic optimal control problems in discrete time with continuous state and action spaces. It proposes a new robust data-driven dynamic programming method, which uses the Nadaraya-Watson kernel-weighted average to approximate conditional expectation. This can also be viewed as using the kernel density estimation to estimate the transition kernel given the training set. Some interesting real-world experiments are provided to demonstrate the superior performance of the algorithm.

My main concern of the paper is on the novelty. In fact, RDDP can be viewed as a special form of robust dynamic programming (see Theorem 2.1 in G. Iyengar [7]). The main contribution of the work is to use the kernel-density estimation for the estimation of the transition probability. This work can be greatly improved if some theoretical justifications can be provided to show that kernel-weighted average is a good choice or establish the sample complexity results. Some detailed comments are as follows:

(1) The authors might wrongly claim that NW estimate is unbiased in Line 146. According to the standard nonparametric estimation theory (see [1]), the bias is bounded by O(h^4) where h is the kernel bandwidth.

(2) Could the authors explain a bit more why the downward bias is the consequence of overfitting ? (It should be underfitting or overfitting?)

(3) In fact, there are many choices for the constraint set for p other than chi-square distance. For example, the KL divergence between q and p. In fact, according to G. Iyengar [7], KL divergence is a lower bound of the Chi-square distances.

(4) For Algorithm 1, how to determine K in real experiments and practice? How about N ?

(5) How to select the prescribed control policy for simulating s ? What polices are used in experiments ?

(6) Minor points: In Appendix, Line 55, it should be "with corresponding optimal value 2 \lambda q_i - 2 q_i \sqrt{\lambda*(\lambda-v_i-\mu)}" instead of "- 2 q_i \sqrt{\lambda /(\lambda-v_i-\mu)}". In addition, some $d$ in $V^d$ should be mathrm font for the consistency (e.g,. Line 140 in main text).

[1] Alexandre B. Tsybakov. Introduction to nonparametric estimation. Springer, 2010.
Summary: This paper proposes a new robust data-driven dynamic programming method, which uses the Nadaraya-Watson kernel-weighted average to approximate conditional expectation. This work could be greatly improved if some theoretical justifications (e.g., sample complexity) can be provided.

Submitted by Assigned_Reviewer_5

Summary:
This paper considers a stochastic optimal control problem in discrete time with continuous state and action spaces, where at any time t, the state of the underlying system has two components (endogenous, exogenous). The endogenous state captures all decision-dependent information, while the exogenous state captures the external random disturbances. Stochastic optimal control problems have many applications in various domains, yet solving these problems via dynamic programming (DP) is challenging due to a number of reasons. In particular, computing backward recursions is burdensome and the distribution of the exogenous states is typically unknown and must be inferred from historical observations, i.e., training data. The main motivation behind this work is to handle the second issue, i.e., when training data is very limited (as compared to the dimension of the problem), the estimated cost-to-go functions display high variability and bias, the resulting control policies perform poorly in out-of-sample tests. To this end, the authors propose a robust data-driven DP approach, where given a set of sample trajectories of the exogenous state, kernel regression is used together with parametric value function approximations to estimate the conditional expectations in DP ([3] shows that the resulting approach is asymptotically consistent). However when the number of sample trajectories is limited, this is still not of much use, due to the noise in the estimates. Therefore, in this paper the authors further propose to replace the expectations in the DP recursions with worst-case expectations over a set of distributions close to the best estimate. The resulting optimization problems are shown to be of conic form and via numerical experiments, they show that new algorithm dominates two non-robust approaches from the literature in two main applications from stock index tracking and wind energy commitment.

Quality:
The proposed data driven approach in this paper is appealing. But I am not convinced that the existing algorithms from the literature are unable to capture the same phenomena. In particular, the authors claim that the robust value iteration methods from the robust Markov decision process (MDP) are not fundamentally data-driven, as their primitives are uncertainty sets for the transition kernels instead of historical observations. Moreover, seemingly the main theoretical contribution of this paper is to show that the resulting optimization problems are conic optimization problems. While arriving at nice conic problems is valuable, I have found no motivation for the Assumption 4.1., which underlies this result. In fact, I find the overall theoretical contribution a bit limited; see my comments in the Originality & Significance part.


I am not convinced that the Assumption 4.1 used is reasonable in a real situation. It seems like this assumption is there to rather make sure that everything fits to the setup of Lemma A.1. Moreover, there is no new algorithmic contribution, as the resulting algorithm is pretty straightforward, and was solved by utilizing the standard solvers SeDuMi and CPLEX. By the way, there is no support offered on how \gamma parameter, which controls the level of robustness desired, should be chosen. Clearly selecting \gamma is very important, as setting it to zero will get us back to the usual data-driven DP approach. Also the brief discussion on this at the beginning of page 5 is not very useful in a real scenario with an unknown distribution.

Finally numerical experiments only demonstrate that this approach is superior to the non-robust standard data-driven DP and the least-squares policy iteration algorithms. None of these methods addresses the robustness issue, a better comparison would be against one of the robust approaches such as robust value iteration method, or actually building uncertainty sets and solving the larger problem described on page 5, lines 234-245, even if it meant imposing additional assumptions on the structure of value functions.

Clarity:
The paper is mostly written in a clear fashion, with a reasonable logical structure, and a lot of examples.

Originality & Significance:
This paper studies an important problem and to the best of my knowledge, the approach is original in robust DP framework. Yet, I believe there is little theoretical contribution in this work. Proof of Lemma A.1, follows trivially from conic duality, note that \triangle(q) =\{p\in\triangle: \exists u s.t. (2(p_i-q_i); u_i-p_i; u_i+p_i )\in\mathcal{L}_3 \forall i=1,…, n\} where \mathcal{L}_3 is the 3 dimensional Lorentz cone, i.e., \mathcal{L}_3=\{x\in\R^3: x_3 \geq \sqrt{x_1^2+ x_2^2}, so \triangle(q) is a conic representable set. Moreover the cones involved are nonnegative orthant and also based on only \ell_1 and \ell_2 norms, so strong conic duality applies here, directly leading to the equivalent formulation given in A.1. I.,e., proof of Lemma A.1, is trivial, in particular, there is no need to use Lagrangian duality. The other proofs for the Theorem and Corollary were already only two lines, following from either Lemma A.1, or previous work. As I stated before, numerical evidence only proves the value of following a robust approach, it does not necessarily prove the value of this particular robust approach.

Summary: I believe this paper studies an important problem and to the best of my knowledge, the approach is original in robust DP framework. Yet, there is little theoretical contribution and I find the numerical results to be of limited use demonstrating the superiority of the approach in the robust DP setup.

Submitted by Assigned_Reviewer_7

This paper introduces a robustness measure into data-driven dynamic programming
account for the inherent variance in the value function estimator. Roughly
speaking, they enforce a minimax guarantee on the performance of their learned
policy with respect to a set of value functions close to their estimate. Under
additional convexity assumptions, they derive a cone program that is used
iteratively to estimate the cost-to-go function.

I recommend that this paper be accepted. The paper is well written and
organized, the exposition relatively easily to follow, and the issues with
existing techniques do not appear to be artificial. Their proposed solution,
though not insightful on its own, develops into a non-trivial algorithm that
appears reasonably practical, and is demonstrated on two real world domains.

I would like to see more discussion regarding the decisions and assumptions
made while deriving RDDP. Though they appear reasonable, it is not immediate
why the alternatives were not considered or how restrictive they are,
respectively. For example:

Presumably alternatives to the Chi-squared distance also lead to equally easy
(convex) optimization problems with a small number of tuning parameters.

How sensitive is the approach to the choice of gamma? or to the
kernel bandwidth parameter?

By allowing the adversary a different choice of p in every subproblem, is the
approach not overly pessimistic?

I would disagree that the convexity assumptions are mild.

The function classes used to approximate the cost-to-go function (piecewise
linear/convex quadratic) seem arbitrary.

The optimization problems scale linearly in the number of sample trajectories.
This seems rather prohibitive. Perhaps integrating an iterative data
collection procedure, like DAgger, could be useful (and by itself may even
alleviate the variance problem).

Comparison against system identification approaches, like PSRs, would be
beneficial.

Minor comments:
- The notation is hard to follow, e.g., V^d_t, I assume the d is for DDP,
but this is never stated.
- Please make the text on the figures larger
Summary: I recommend this paper be accepted. It is clearly written and solves a problem with existing techniques.
Author Feedback

Author rebuttal: We sincerely thank the reviewers for their appreciation of our results and their thoughtful comments. We would like to offer the following clarifications on the questions raised in the reports.

Referee 4 remarks that RDDP can be viewed as a special form of robust DP due to Iyengar [7]. We agree with this observation but would also like to highlight the following statement from Iyengar’s paper: “We restricted our attention to problems where the non robust DP is tractable. In most of the interesting applications of DP, this is not the case and one has to resort to approximate DP.” Our paper tries to address this point. We propose a systematic way to formulate and solve intractable robust DP problems with continuous state and action spaces, and we showcase the benefits of a robust approach in different real-life applications based on (sparse) real data. For these problems, to the best of our knowledge, robust value iteration would require a discretization of the state and action spaces and would therefore scale exponentially in the problem dimensions (our wind energy commitment problem involves state and action spaces with 51 and 168 dimensions per time period, respectively).

Referee 5 correctly points out that existing algorithms for solving robust MDPs (e.g. the one by Wiesemann et al.) can also mitigate overfitting phenomena. However, most of these methods assume finite state and action spaces and are therefore limited to low-dimensional problems that can be discretized. Also, the underlying historical observations are drawn from an atomic probability space and are therefore inherently different from the non-atomic ones considered in our paper. Moreover, to the best of our knowledge, all of them are tested only on simulated data. We thank referee 5 for the pointer to the paper by Wiesemann et al., which we will cite in the revision along with other recent contributions to the literature on robust MDPs.

Referee 5 points out that Theorem 4.1 and Lemma A.1 are easy to prove. We completely agree, and we actually see this as a benefit. RDDP is easy to grasp and implement, and our numerical results based on real data suggest that it works well at least in some application domains. Thus, there is hope that RDDP might be adopted by non-experts with limited knowledge of robust optimization.

Referees 5 and 7 question the reasonableness of Assumption 4.1. We agree that the assumption is restrictive even though it is satisfied by many interesting problems. However, the assumption can be significantly relaxed to allow for nonconvexities in the endogenous state. There is a brief comment on this in the conclusions section. We will make this clearer in the revision.

We would like to thank Referee 4 for the reference on nonparametric estimation, which we will cite. As the small bias from NW estimation is dominated by the large bias introduced through the optimization, we believe that our heuristic argument on page 4 is valid. The downward bias in V^{d}_{t} is caused by overfitting as the optimization in (4) can exaggerate fluctuations in the samples. In Example 3.1, for instance, the permutation symmetry of the true optimal solution is broken in DDP unless we draw (by coincidence) a symmetric sample set.

Referee 4 asks what policy is used to simulate the endogenous state trajectories. If the problem is of a constrained LQR form, we can use the exact optimal policy of the corresponding relaxed LQR problem. If the endogenous state has low dimension (e.g., <4), the evaluation points s_{t}^{k} can be sampled uniformly from the set of feasible endogenous states (such an approach is also used in [5] and [26]). In all other cases, we use a naive model predictive control policy.

Referee 7 requests information on the choice of the model parameters:
- We choose K large enough such that the approximate value function at the sample points does not change significantly (w.r.t. the l_2-norm) when new samples are added.
- N can be selected in a similar manner as K. However, in practice we typically use all available historical observations as N is assumed to be small.
- We use the chi-squared distance to construct uncertainty sets as it assigns nonzero weights to all samples (which prevents our model from being overly pessimistic) and because the corresponding worst-case expectation problem is equivalent to an explicit SOCP. Uncertainty sets based on the KL divergence, for instance, would lead to convex optimization problems that still admit self-concordant barrier functions but are less tractable than SOCPs.
- gamma is determined via cross-validation. In our experience the solutions of RDDP are rather insensitive to gamma. This is also a desirable consequence of using a chi-squared uncertainty set, which rules out degenerate weights that assign all mass to a single observation.

We will clarify all of the above points in the revision.

We would like to thank the reviewers again for their valuable feedback!